# Sodium Butyrate as Key Regulator of Mitochondrial Function and Barrier Integrity of Human Glomerular Endothelial Cells

**DOI:** 10.3390/ijms241713090

**Published:** 2023-08-23

**Authors:** Maria Novella Nicese, Roel Bijkerk, Anton Jan Van Zonneveld, Bernard M. Van den Berg, Joris I. Rotmans

**Affiliations:** 1Department of Internal Medicine, Division of Nephrology, Leiden University Medical Center, Albinusdreef 2, 2333 ZA Leiden, The Netherlands; m.n.nicese@lumc.nl (M.N.N.); r.bijkerk@lumc.nl (R.B.); a.j.van_zonneveld@lumc.nl (A.J.V.Z.); b.m.van_den_berg@lumc.nl (B.M.V.d.B.); 2Einthoven Laboratory for Vascular and Regenerative Medicine, Leiden University Medical Center, Albinusdreef 2, 2333 ZA Leiden, The Netherlands

**Keywords:** glomerular endothelial cells, short chain fatty acids, butyrate, proliferation, endothelial barrier, mitochondria, seahorse, LPS

## Abstract

The gut microbiota has emerged as an important modulator of cardiovascular and renal homeostasis. The composition of gut microbiota in patients suffering from chronic kidney disease (CKD) is altered, where a lower number of bacteria producing short chain fatty acids (SCFAs) is observed. It is known that SCFAs, such as butyrate and acetate, have protective effects against cardiovascular diseases and CKD but their mechanisms of action remain largely unexplored. In the present study, we investigated the effect of butyrate and acetate on glomerular endothelial cells. Human glomerular microvascular endothelial cells (hgMVECs) were cultured and exposed to butyrate and acetate and their effects on cellular proliferation, mitochondrial mass and metabolism, as well as monolayer integrity were studied. While acetate did not show any effects on hgMVECs, our results revealed that butyrate reduces the proliferation of hgMVECs, strengthens the endothelial barrier through increased expression of VE-cadherin and Claudin-5 and promotes mitochondrial biogenesis. Moreover, butyrate reduces the increase in oxygen consumption induced by lipopolysaccharides (LPS)*,* revealing a protective effect of butyrate against the detrimental effects of LPS. Taken together, our data show that butyrate is a key player in endothelial integrity and metabolic homeostasis.

## 1. Introduction

Chronic kidney disease (CKD) is a progressive disease affecting over 800 million people worldwide, with a high prevalence in patients with hypertension and type 2 diabetes mellitus (T2DM) [1]. CKD is characterized by glomerular injury [2], with the endothelial barrier being one of its main components. [3]. In the case of CKD, a high inflammatory state, increased oxidative stress levels and impaired nitric oxide (NO) production can damage the glomerular filtration system and cause endothelial dysfunction (ED) [4]. ED is characterized by a loss of cell-to-cell contacts which, together with an increased production of platelet-derived growth factor β (PDGF-β) and vascular endothelial growth factor A (VEGFA), promotes vascular instability and rarefaction [5,6,7,8]. This loss in endothelial integrity in CKD is also accompanied by altered mitochondrial function [9]. While mitochondrial homeostasis is regulated by cycles of fission–fusion and mitophagy–mitogenesis [10], this balance is skewed towards fission in T2DM and hypertension-driven hypoxia. This phenomenon causes excessive mitochondrial fragmentation, leading to mitochondrial dysfunction and a progressive reduction in the number of mitochondria [11,12].

The gut microbiota has recently emerged as a modulator of CKD pathology [13]. In CKD patients, the concentrations of microbiota-derived trimethylamine N-oxide (TMAO), indoxyl sulfate and p-cresyl sulfate in the blood are increased [14]. Because of poor renal function in CKD patients, these uremic toxins remain in the circulation, promote inflammation, thereby contributing to progressive kidney injury [15]. Moreover, these metabolites directly contribute to the microbiome imbalance by boosting uremic toxin-producing bacteria and lowering the ones producing short chain fatty acids [16,17]. Short chain fatty acids (SCFAs), including butyrate and acetate, are associated with beneficial effects on human health [18,19,20]. Previous studies showed that SCFAs also exert anti-inflammatory properties in cultured endothelial cells, as they inhibit leukocyte adhesion [21] and reduce expression of the pro-inflammatory proteins ICAM-1 and VCAM-1 [22]. Moreover, it has been reported that butyrate can reduce oxidative stress by improving mitochondrial function in STZ-treated pancreatic β-cells [23] and neuronal cells [24]. Butyrate is also able to counteract the pro-inflammatory activity of gut-derived lipopolysaccharides (LPS) [25], which, besides triggering inflammation and oxidative stress [26], promotes an excessive intestinal permeability [27]. Interestingly, CKD patients display an increased amount of LPS in both intestine and blood [28]. 

While previous studies showed that SCFAs can have protective effects on CKD progression [13,29], the direct effects of these metabolites on the glomerular endothelium remain still elusive. For this reason, we investigated the effect of butyrate and acetate on the function and metabolism of human glomerular microvascular endothelial cells (hgMVECs).

## 2. Results

### 2.1. Butyrate Decreases Proliferation of hgMVECs

After exposing cells to increasing concentrations of butyrate or acetate for either 24 h or 48 h, cells were stained for ki-67 to identify proliferating cells. After 24 h incubation, butyrate significantly reduced proliferation of hgMVECs at every concentration tested (Figure 1A,B), unlike acetate (Figure 1C,D). After 48 h, only butyrate at 500 μM was effective in reducing proliferation of hgMVECs (Figure 1E,F).

### 2.2. Butyrate Improves Endothelial Monolayer Integrity by Increasing the Expression of Proteins Involved in Endothelial Junction Formation

To assess the impact of the SCFAs on the endothelial monolayer resistance, cells were treated with either butyrate or acetate for at least 24 h after the formation of a stable monolayer. Butyrate at 250 μM and 500 μM increased the resistance of the endothelial monolayer (Figure 2A,B), while acetate did not show any effect on the resistance of the monolayer at any of the concentrations tested (Figure 2C,D). 

To visualize cell-to-cell contact, cells were fixed and stained for claudin-5 and VE-cadherin after 48 h exposure with butyrate or acetate. We observed that butyrate at 250 μM and at 500 μM increased the expression of both claudin-5 (Figure 2E,F) and VE-cadherin (Figure 2H,I). Acetate did not influence the expression of these proteins at any of the concentrations tested (Figure 2E,G,H,J).

### 2.3. Butyrate Increases Mitochondrial Mass and Influences the Expression of Genes Involved in Mitochondrial Homeostasis in hgMVECs

To visualize the effect of butyrate and acetate on mitochondrial mass and mitochondrial membrane potential, MitoTracker™ Green and MitoTracker™ Deep Red were used, respectively (Figure 2A,B). Upon 24 h-exposure to butyrate, the mitochondrial mass in hgMVECs was significantly increased, especially at the concentrations of 100 μM and 500 μM (Figure 2C). In contrast, acetate did not show the same effect on the mitochondrial mass at any of the concentrations tested (Figure 2D). While affecting the mitochondrial mass, butyrate did not influence mitochondrial membrane potential, nor did acetate (Figure 2E,F). 

To assess the effect of butyrate and acetate on the expression of genes involved in mitochondrial metabolism, qPCR was performed on genes involved in mitochondrial fusion (OPA1, MFN1, MFN2) fission (DRP1, FIS1), mitophagy (PINK1), and mitochondrial biogenesis (PGC1-α). Both butyrate (Figure 3G) and acetate (Figure 3H) downregulated genes involved in mitochondrial fusion, fission and mitophagy. When exposing to butyrate, we were able to detect PGC1-α gene expression, which was expressed at a low level or not detected in control conditions. When exposing cells to acetate, expression of PGC1-α was not detected. 

### 2.4. Butyrate Decreases Maximal Respiratory Capacity of hgMVECs 

To study the effect of SCFAs on mitochondrial function, hgMVECs were exposed to butyrate or acetate for 24 h. Then, mitochondrial function was studied using a seahorse assay, specifically assessing the oxygen consumption rate. Butyrate decreased the maximal respiration of hgMVECs in a dose-dependent manner (Figure 4A,D). On the other hand, acetate did not show the same type of effect (Figure 4E,H). Unlike maximal respiration, basal respiration and ATP-linked respiration of hgMVECs were not significantly influenced by either butyrate (Figure 4B,C) or acetate (Figure 4F,G).

### 2.5. Butyrate Reduces the Increased Maximal Respiratory Capacity Induced by LPS

To study the effect of butyrate and acetate on mitochondrial function in combination with LPS, hgMVECs were exposed for 24 h to either butyrate or acetate and LPS. Exposure to LPS caused a general increase in OCR (Figure 5A,E), with a significant increase in basal respiration and maximal respiratory capacity (Figure 5B,D,F,H). Co-incubation of LPS with butyrate led to a significant decrease in the maximal respiratory capacity and in general to a lower OCR (Figure 5B–D). On the other hand, acetate did not show the same type of effect (Figure 5F–H). Although OCR seems to decrease when administering 500 μM of acetate, the difference when comparing to vehicle + 100 ng/mL LPS is not statistically significant. 

## 3. Discussion

In the present study, we explored the effect of butyrate and acetate on human glomerular microvascular endothelial cells (hgMVECs). Specifically, we aimed to understand whether these compounds affect the homeostasis of glomerular endothelial cells and whether SCFAs might play a protective role against endothelial dysfunction (ED). 

### 3.1. Butyrate Inhibits Endothelial Cell Proliferation

One of the main features of ED, especially in diabetic conditions, involves the uncontrolled proliferation of endothelial cells [30], which is mainly caused by an unbalanced production of VEGFA and NO. The latter helps to control proliferation of glomerular endothelial cells, but in diabetic conditions the producing enzyme, endothelial NO synthase (eNOS), is impaired. This leads to a lower production of NO which in combination with an increased production of VEGFA by podocytes causes uncontrolled proliferation [31,32]. In our study, we showed that butyrate can reduce proliferation of glomerular endothelial cells and potentially keep proliferation under control. This action of butyrate can be due to its ability to restore NO production through GPR41/43 activation [33] but also to its epigenetic activity. Butyrate is a well-known and potent histone deacetylase (HDAC) inhibitor and in this way, it can influence the expression of genes, also of the ones involved in cellular proliferation. For instance, a previous study showed that butyrate decreases the expression of the proliferative genes TES and PGF through HDC inhibition [34]. At the same time, SCFAs can have direct metabolic effects as they enter and boost the tricarboxylic (TCA) cycle, while reducing glycolysis. As a consequence, the proliferation of endothelial cells may also be reduced [35].

### 3.2. Butyrate and Increases the Resistance of the Endothelial Monolayer

The ability of butyrate to modulate the genetic profile of hgMVECs is evident also when looking at the expression of proteins involved in cell-to-cell adhesion. Treatment with butyrate resulted in the upregulated expression of VE-cadherin and claudin-5 and consequently in an increased resistance of the endothelial monolayer. The integrity of the endothelial monolayer is crucial for the functioning of any type of vessel, so that the proper degree of permeability is guaranteed. However, stressors such as hypoxia or oxidative stress can influence vascular permeability by hampering barrier function and integrity. For instance, exposure of endothelial cells to hydrogen peroxide promotes a loss of occludin and cadherin [36], while a previous study revealed that monolayer resistance of endothelial cells was impaired after exposure to a pro-inflammatory stimulus. Interestingly, barrier integrity was then restored upon incubation with vitamin D, which coincided with increased expression of VE-cadherin [37]. In a similar way, inflammation and oxidative stress can also hamper the integrity of the blood–brain barrier (BBB) via an increased activation of the Nf-kΒ/p65 pathway. This leads to a lower expression of claudin-5 and therefore to a weakened BBB. However, it has been shown that sodium butyrate can reduce the activity of the Nf-kΒ/p65 pathway and in this way restore the expression of claudin-5 [38].

Therefore, the increase in expression of VE-cadherin and also claudin-5 induced by butyrate suggest that this type of short chain fatty acid can play a pivotal role in protecting the integrity of the renal endothelial barrier against ED.

### 3.3. Butyrate Regulates Mitochondrial Biogenesis and Mitochondrial Function

As previously mentioned, excessive mitochondrial fragmentation and loss is one of the main features underlying mitochondrial dysfunction and, ultimately, kidney injury [39]. Interestingly, in our study, we observed a clear upregulation of the mitogenesis regulator PGC1-α upon butyrate exposure, which coincided with an increase in mitochondrial mass. Importantly, when staining with MitoTracker^TM^ Deep red, we could also observe that the increase in mitochondrial mass was not followed by a decrease in mitochondrial membrane potential, suggesting that this is not disturbed by butyrate. However, the higher number of mitochondria did not correspond to an augmented mitochondrial function, which declined instead in a dose-dependent manner. This might be explained by the action of butyrate as an HDAC inhibitor and by the presence of a potential negative feedback mechanism. When the ratio of acetyl-CoA/CoA increases excessively, acetyl-CoA has been shown to inhibit the activity of pyruvate dehydrogenases and consequently the amount of substrate available for mitochondrial respiration [40]. 

This could also partially explain how butyrate is able to restore mitochondrial function upon LPS treatment. LPS significantly increased basal respiration and maximal respiration of hgMVECs, which is necessary to provide energy to support protein and lipid production when endothelial cells are activated [41]. These cells respond to LPS stimulation by increasing mitochondrial respiration through the inhibition of the forkhead box protein O1/pyruvate dehydrogenase kinase 4 (FOXO1/PDK4) pathway, which regulates the amount of pyruvate that can enter the TCA cycle [42]. Interestingly, fatty acid supplementation seems to upregulate PKD4 [43] and this action could also contribute to the reduction in OCR when co-incubating butyrate and LPS. Taken all together, these observations suggest that butyrate can protect glomerular endothelial cells against mitochondrial dysfunction.

Unlike butyrate, acetate did not exert an effect on hgMVECs at any of the concentrations tested. One of the possible explanations underlying these observations is that butyrate is considered to be a more potent HDAC inhibitor compared to acetate [44] but also that butyrate and acetate have a different metabolic turnover. For instance, it is known that acetate shows a higher affinity for acyl-CoA short-chain synthetases (ACSS) than butyrate, and this induces a more rapid conversion of acetate in acetyl-CoA, when compared to butyrate [45]. 

### 3.4. Possible Clinical Implications

Our data suggest that butyrate plays a pivotal role in glomerular endothelial homeostasis, which has not only cellular but also potential clinical implications. With respect to CKD for instance, a recent study [16] showed that fecal and serum SCFAs are remarkably higher in healthy controls than in patients affected by CKD. Moreover, the progression of CKD negatively correlates with the amount of butyrate detected in the serum of the patients. This suggests that butyrate supplementation might have potential beneficial effects on kidney function, as it has been illustrated in several animal studies [46,47,48]. Besides helping to preserve kidney function, SCFAs seem to be beneficial for the whole cardiovascular system, by showing anti-hypertensive effects [49] and protecting from atherosclerosis [50] and heart failure [51]. Interestingly, the benefits of butyrate also go beyond the cardiovascular system, as they have been observed for instance acting on the central nervous system [52] and on intestinal barrier integrity [53,54]. The latter, in particular, is pivotal in preventing toxins produced in the gut from reaching other organs through the circulation. Importantly, butyrate administration might also have a long-term positive effect since it is able to reshape the gut microbiota composition and progressively boost and replenish SCFAs bacteria [55]. 

Although most of the studies have been conducted on cells and animals so far, all the gathered data suggest that butyrate supplementation might be an important ally in preventing the onset and the progression of cardiovascular and other severe diseases.

### 3.5. Study Limitations

The main limitation of this study is that cells have been treated with butyrate and acetate in a healthy setup, except for the experiments involving LPS. For this reason, in the future we are planning to study the effect of butyrate in a disease-mimicking environment, for instance by pre-incubating glomerular endothelial cells with diabetic serum and oxidative stress inducers. In this way we aim to further characterize the mechanisms of action of SCFAs on hgMVECs in endothelial dysfunction and CKD. 

In conclusion, in our study we showed that in human glomerular endothelial cells butyrate decreases cellular proliferation and promotes mitochondrial biogenesis as well as monolayer integrity. Moreover, butyrate can counteract the respiration burst induced by LPS, suggesting a protective effect of butyrate against inflammation. Therefore, our data indicate that butyrate might play a pivotal role in preventing the onset and progression of endothelial dysfunction in CKD. 

## 4. Materials and Methods 

### 4.1. Primary Human Glomerular Microvascular Endothelial Cells

Primary human glomerular microvascular endothelial cells (hgMVECs) were purchased at Cell-System (Kirkland, WA, USA; ACBRI-128). Cells were cultured at 37 °C with 5% CO_2_ in full endothelial growth medium-2 (EGM-2) supplemented with 100 IU/mL penicillin and 100 μg/mL streptomycin. For the experiments, cells were left attached for 24 h and then treated for 24 h or 48 h with sodium butyrate (Sigma, St. Louis, MI, USA, 303410) or sodium acetate (Sigma, St. Louis, MI, USA; S2889-250G), or for 24 h with sodium butyrate/sodium acetate and 100 ng/mL of lipopolysaccharides from E. Coli O111:B4 (Sigma, St. Louis, MI, USA; L4391).

### 4.2. Ki-67 Staining for Proliferating Cells

Cells were stained with a monoclonal antibody against ki-67 (556003; BD-Pharmingen, San Diego, CA; USA) to visualize proliferating cells. First, cells were fixated with a solution at 4% of paraformaldehyde (PFA) for 10 min at room temperature. Afterwards, blocking and permeabilization steps were performed with a solution of 0.3% Triton-1% BSA for 10 min. Cells were then incubated with the primary antibody against ki-67 for 1.5 h at room temperature. After washing with PBS, cells were incubated with secondary antibody (Invitrogen, Waltham, MA, USA; A-11001) and stained with Hoechst (Thermo Fisher, Waltham, MA, USA; H3569) to visualize the nuclei for 45 min. After washing with PBS, coverslips were mounted with ProLong Gold (Thermo Fisher, Waltham, MA, USA; P36930) and cells were visualized using fluorescent microscopy. 

### 4.3. Mitochondrial Staining with MitoTracker™

MitoTracker™ Green (Thermo Fisher, Waltham, MA, USA; M7514) and MitoTracker™ Deep Red (Thermo Fisher, Waltham, MA, USA; M22426), were used to stain for mitochondrial mass and mitochondrial membrane potential, respectively. Cells were incubated with 25 nM of MitoTracker™ Green or MitoTracker™ Deep Red for 30 min at 37 °C. Afterwards, cells were washed with warm medium and directly imaged with confocal microscopy. 

### 4.4. Seahorse Assay

To study mitochondrial metabolism, seahorse assay experiments were performed. For this assay, cells were cultured in a Seahorse XF96 polystyrene tissue culture plate (Agilent technology, Santa Clara, CA, USA; 102416-100) coated with 1% gelatin. Cells were incubated with acetate or butyrate only, or with acetate/butyrate and LPS. Prior to the run, cells were incubated for 1 h at 37 °C without CO_2_ with Seahorse XF base medium (Agilent technology, Santa Clara, CA, USA; 103334-100) supplemented with L-glutamine. To measure oxygen consumption rate (OCR), we used the following compounds during the assay: 10 mM glucose (Thermo Fisher, Waltham, MA, USA; 15023021), 5 μM oligomycin A (Cayman Sanbio, Tokyo, Japan; 11342), 1 μM FCCP (Sigma, Waltham, MA, USA; C2920), 1 μM antimycin a (Sigma, Waltham, MA, USA; A8674) and 1 μM rotenone (Sigma, Waltham, MA, USA; R8875). Once the run terminated, cellular protein content was determined with a Pierce™ BCA-protein kit (Thermo Fisher, 23225). OCR values were then normalized to the protein content of the corresponding well. 

### 4.5. Barrier Function Assay

Endothelial barrier resistance was analyzed by using an electric cell-substrate impedance sensing system (ECIS, Singapore, Applied BioPhysics). Cells were cultured on a ECIS Cultureware 96W20idf PET (Ibidi, Fitchburg, WI, USA; 72098) previously coated with 10 mM of L-cysteine, and then 1% gelatin. When cells were in the plate for 24 h and reached baseline resistance, butyrate or acetate were added to the plate. 

### 4.6. Staining for VE-Cadherin and Claudin-5

Cells were stained with antibodies against VE-cadherin (Pharmingen, San Diego, CA, USA; 55561) and claudin-5 (Invitrogen, Waltham, MA, USA; 35-2500) to study the expression of these proteins as markers of endothelial junction formation. After incubating the cells with butyrate or acetate, we performed a fixation step with 4% PFA and a permeabilization-blocking step with 0.3% triton-1% BSA. Cells were then incubated with the primary antibody against either VE-cadherin or claudin-5 for 1.5 h at room temperature. After washing with PBS, cells were incubated with secondary antibody (Invitrogen, Waltham, MA, USA; A-11001) and stained with Hoechst (Thermo Fisher, Waltham, MA, USA; H3569) to visualize the nuclei for 45 min. After washing with PBS, cells were ready to be imaged with fluorescence microscopy.

### 4.7. Quantitative PCR

RNA was isolated by using an RNeasy Mini Kit (Qiagen, Singapore, 74106) according to manufacturer’s instructions. cDNA was then produced from the RNA using a M-MLV Reverse Transcriptase Kit (M1701; Promega, Singapore), while qPCR was performed with SYBR Select Master Mix (4472908, Waltham, MA, USA; Applied Biosystems). Forward and reverse primers of the following genes were used to perform qPCR analysis (Table 1). 

### 4.8. Analysis and Statistics

Analysis of the images was performed with ImageJ-win64 (Fiji). The results are shown as mean ± SEM, while *n* represents the number of biological independent experiments. Differences between groups were assessed with One-Way ANOVA, where *p* values of <0.05 were considered statistically significant. Specifically, one star is used to indicate *p* values between 0.05 and 0.01, two stars for *p* values between 0.01 and 0.001 and three stars for *p* values between 0.001 and 0.0001.

## Figures and Tables

**Figure 1 ijms-24-13090-f001:**
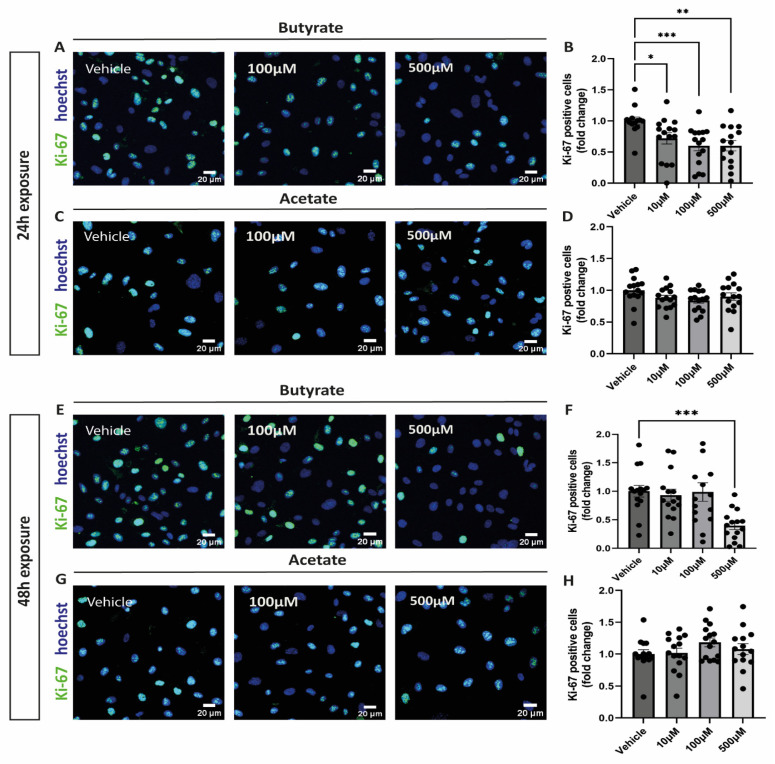
Effect of butyrate and acetate on proliferation marker ki-67 in hgMVECs. hgMVECs have been exposed to either acetate or butyrate for 24 h (**A**–**D**) or 48 h (**E**–**H**). For the analysis, we counted the total number of nuclei (blue) in every picture and the percentage of ki-67 positive nuclei (green). The results are plotted as the fold change compared to vehicle. For every experiment (*n* = 3), we took 5 random pictures per condition, that were used for quantification. Data are presented as mean ± SEM. Statistical significance was analyzed by using One-Way ANOVA analysis, with * for *p* values < 0.05, ** for *p* values < 0.01 and *** for *p* values < 0.001.

**Figure 2 ijms-24-13090-f002:**
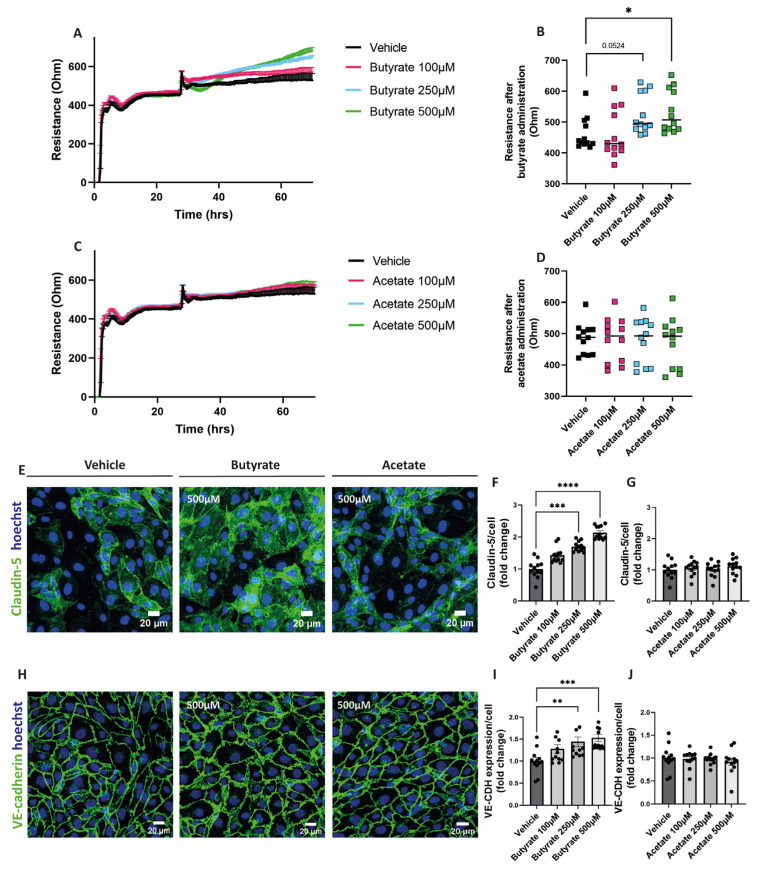
Effect of butyrate and acetate on monolayer resistance and expression of claudin-5 and VE-cadherin in hgMVECs. For the ECIS experiments, hgMVECs were exposed to either butyrate (**A**) or acetate (**C**) for at least 24 h. For the analysis, we considered the time after we applied the treatment either with butyrate (**B**) or acetate (**D**). The data collected from every experiment (*n* = 3) are presented as mean ± SEM. For the stainings of claudin-5 (**E**–**G**) and VE-cadherin (**H**–**J**), hgMVECs were exposed to either butyrate or acetate for 48 h. For every experiment (*n* = 3), 4 random pictures per well were used for quantification purposes. The results of every experiment (*n* = 3) are plotted as the fold change compared to the vehicle. Data are presented as means ± SEM. Statistical significance was analyzed by using One-Way ANOVA analysis, with * for *p* values < 0.05, ** for *p* values < 0.01, *** for *p* values < 0.001 and **** for *p* values < 0.0001.

**Figure 3 ijms-24-13090-f003:**
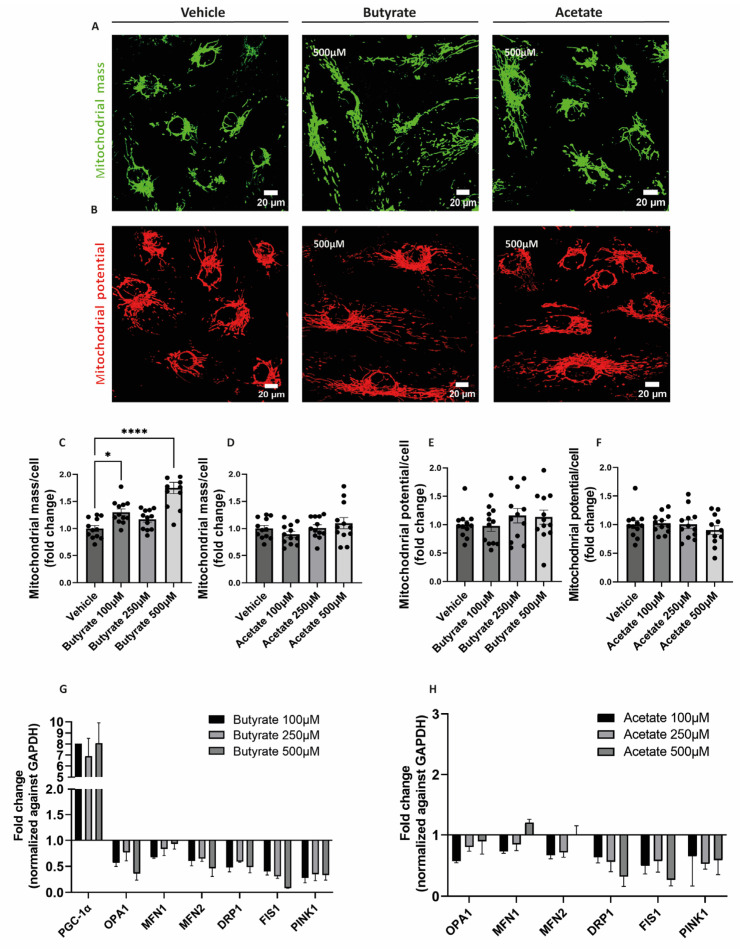
Effect of butyrate and acetate on mitochondrial mass, mitochondrial membrane potential and gene expression of mitochondrial genes of hgMVECs. hgMVECs were exposed to butyrate or acetate for 24 h before staining with MitoTracker™ Green (**A**) or MitoTracker™ Deep Red (**B**). For the analysis of mitochondrial mass, we calculated the mitochondrial area, which was then normalized by the number of cells (**C**,**D**). For the mitochondrial potential, we calculated the intensity of the signal per cell (**E**,**F**). The results for both mitochondrial mass and mitochondrial potential were then plotted as fold change compared to the vehicle. For every experiment (*n* = 3) we took 4 random pictures per well, which were then used for quantification. For the qPCR experiments we collected mRNA from different experiments (*n* = 3) where cells were exposed to either butyrate or acetate for 24 h. The results of every experiment are plotted as the fold change compared to the control after normalization against GAPDH (**G**,**H**).The results of every experiment are plotted as the fold change compared to the vehicle. Data are presented as mean ± SEM. Statistical significance was calculated by using One-Way ANOVA analysis, with * for *p* values < 0.05 and **** for *p* values < 0.0001.

**Figure 4 ijms-24-13090-f004:**
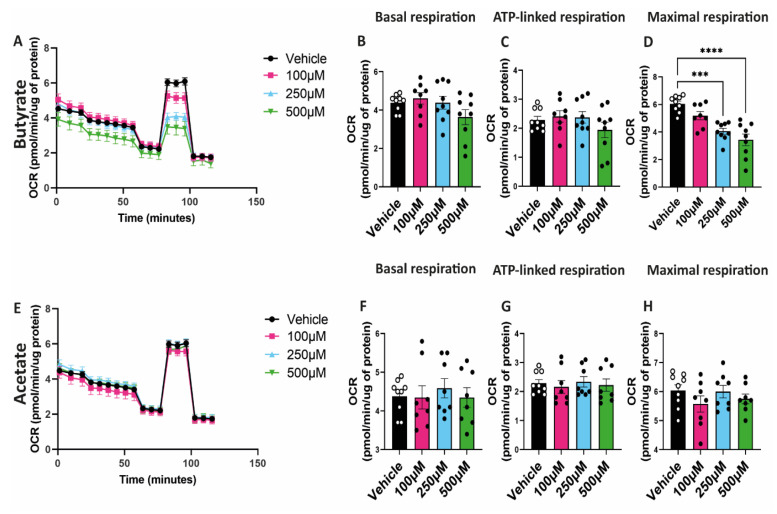
Effect of butyrate and acetate on OCR of hgMVECs. hgMVECs were exposed to butyrate (**A**–**D**) or acetate (**E**–**H**) for 24 h before performing the seahorse assay. For every experiment (*n* = 3) we calculated basal respiration, ATP-linked respiration and maximal respiration. Data are presented as mean ± SEM. Statistical significance was calculated by using One-Way ANOVA analysis, with *** for *p* values < 0.001 and **** for *p* values < 0.0001.

**Figure 5 ijms-24-13090-f005:**
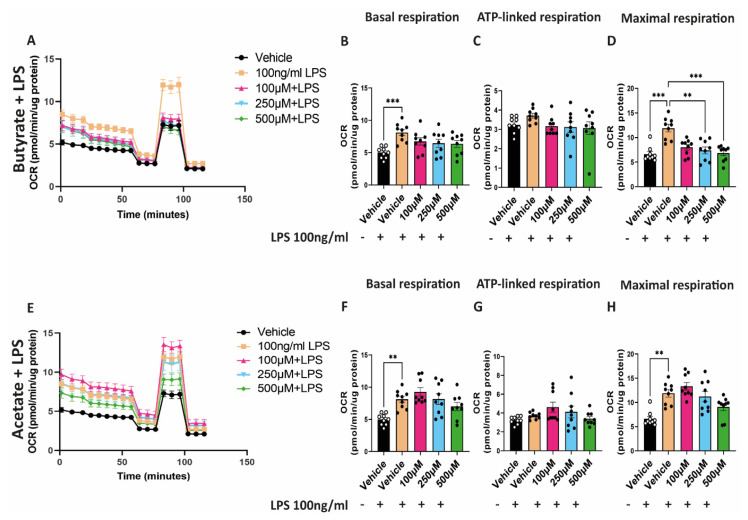
Effect of butyrate (**A**–**D**) and acetate (**E**–**H**) in combination with LPS on OCR of hgMVECs. hgMVECs were co−incubated with butyrate or acetate and LPS for 24 h before performing seahorse assay. For every experiment (*n* = 3) we calculated basal respiration, ATP−linked respiration and maximal respiration. Data are presented as mean ± SEM. Statistical significance was calculated by using One-Way ANOVA analysis, with ** for *p* values < 0.01, *** for *p* values < 0.001.

**Table 1 ijms-24-13090-t001:** Primer sequences for qPCR analysis.

Gene Name	Forward Sequence 5′→3′	Reverse Sequence 3′→5′
*DPR1*	TTCCATTATCCTCGCTGTCAC	CATCAGTACCCGCATCCATG
*FIS1*	TGACATCCGTAAAGGCATCG	CTTCTCGTATTCCTTGAGCCG
*MFF*	TAAATGAGTAAAGGAACAAGCAGTG	AGCAGTGGGAGAAGGAAATG
*PGC-1α*	CAGGCAGTAGATCCTCTTCAAG	TCCTCGTAGCTGTCATACCTG
*PINK*	GAGTATGGAGCAGTCACTTACAG	CAGCACATCAGGGTAGTCG
*VE-cadherin*	GCACCAGTTTGGCCAATATA	GGGTTTTTGCATAATAAGCAGG
*Claudin-5*	GTTTTACGACCCGTCTGTGC	AGTGGCAGGAGAAGGTCAGC
*GAPDH*	CCAGGCGCCCAATACG	CCACATCGCTCAGACACCAT

## Data Availability

The data presented in this study are available on request from the corresponding author.

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
