# Peer review of "Sodium Butyrate as Key Regulator of Mitochondrial Function and Barrier Integrity of Human Glomerular Endothelial Cells"

_ijms, 2023, doi:10.3390/ijms241713090_

Round 1

Reviewer 1 Report

Very carefully prepared manuscript by Dutch authors. They showed that butyrate is a cardinal player in endothelial integrity and metabolic homeostasis.

Here are the notes to the manuscript:

1. the abstract and introduction are well-written and the purpose is carefully formulated.

2. the results are clearly formulated and support the conclusions given by the author.

3. in the discussion, the authors should present the potential clinical significance of their results.

4. what are the limitations of the study?

5. potential manuscripts for discussion are:

https://www.mdpi.com/2311-5637/9/3/205

https://www.mdpi.com/1420-3049/27/24/8715

https://www.mdpi.com/2073-4409/12/4/657

https://www.mdpi.com/2072-6643/15/4/930

https://www.mdpi.com/2079-7737/12/2/145

Reviewer 2 Report

The subject is interesting, the number of experiments is adequate for supporting the conclusions, the manuscript is well-written, and a large quantity of information is presented concisely.

In my opinion, the manuscript is suitable for publication.

I only have one minor comment. It would be interesting for the authors to provide their views about the possible therapeutic implications of their findings. Is it possible to alter CKD patients' gut microbiome so that more butyrate can be produced? Can butyrate be administered?  

Round 2

Reviewer 1 Report

The authors have addressed all my questions and concerns.